# Pharmaceutical Company Targets and Strategies to Address Climate Change: Content Analysis of Public Reports from 20 Pharmaceutical Companies

**DOI:** 10.3390/ijerph20043206

**Published:** 2023-02-11

**Authors:** Amy Booth, Alexandra Jager, Stuart D Faulkner, Christopher C Winchester, Sara E Shaw

**Affiliations:** 1Nuffield Department of Primary Care Health Sciences, University of Oxford, Oxford OX2 6GG, UK; 2Oxford PharmaGenesis, Oxford OX13 5QJ, UK

**Keywords:** climate change, greenhouse gas emissions, pharmaceutical industry, corporate social responsibility

## Abstract

The pharmaceutical industry produces a large proportion of health system greenhouse gas (GHG) emissions, contributing to climate change. This urgently needs to be addressed. We aimed to examine pharmaceutical company climate change targets, GHG emissions, and strategies to reduce them. We performed content analysis of the 20 largest pharmaceutical companies’ publicly available 2020/2021 reports, focusing on extracting information on their reported climate change targets, GHG emissions (and whether companies had demonstrated any reduction in emissions over their reporting period), and strategies being implemented to reduce company emissions and meet their targets. Nineteen companies have committed to reducing GHG emissions, ten to carbon neutrality and eight to net zero emissions between 2025 and 2050. Companies showed largely favorable reductions in scope 1 (in-house) and scope 2 (purchased energy), with variable results in scope 3 (supply chain) emissions. Strategies to reduce emissions included optimizing manufacturing and distribution, and responsible sourcing of energy, water, and raw materials. Pharmaceutical companies are setting climate change targets and reporting reduced emissions via a range of strategies. This varies, with scope to track actions and accountability to targets, improve consistency of reporting, especially of scope 3 emissions, and collaborate on novel solutions. There is need for further mixed methods research on progress with achieving reported climate change targets, as well as implementation of strategies to reduce emissions within the pharmaceutical industry.

## 1. Introduction

Health systems produce 4–5% of national greenhouse gas (GHG) emissions [1], contributing to climate change, with its subsequent negative impact on human health [2]. Health systems are starting to set stringent targets to reduce GHG emissions; for example, the UK National Health Service (NHS) has committed to net zero across its operations and supply chain by 2045 [3]. These targets can only be achieved with support from health system suppliers, such as pharmaceutical companies, who contribute a significant proportion of health sector GHG emissions. In 2019, 20% of the carbon footprint of the NHS was due to medicines and chemicals and an additional 5% was due to anesthetic gases (2%) and inhalers (3%) [4]. In other national estimates, the contribution of pharmaceuticals to health systems GHG emissions ranges from 10–55% [1,5]. In addition to their responsibilities to customers in health systems, pharmaceutical companies have corporate social responsibilities to their investors and wider society that are increasingly including setting climate change targets and reporting GHG emissions [6].

### 1.1. Terminology

Climate change targets typically focus on GHG emission reduction. The terminology used to encompass these targets is broad, but includes, for example, pledges to carbon neutrality, net zero, or similar commitments (e.g., a percentage reduction in greenhouse gas emissions over a certain time period). The range of terms used can be found in Box 1.

Box 1Climate change terminology (adapted from Intergovernmental Panel on Climate Change Glossary [7]).Climate neutral: State in which human activities have no net effect on the climate system.Climate positive: State in which human activities result in beneficial effects on the climate system.Climate target: temperature limit, concentration level, or emissions reduction goal used with the aim of avoiding anthropogenic
(human-caused) impact on the climate system.Greenhouse gas emissions: natural and anthropogenic gases (e.g., carbon dioxide, nitrous oxide, methane) that absorb and emit radiation
causing heat to be trapped within the atmosphereNegative emissions: Deliberate removal of GHGs from the atmosphere by human activities.Net negative emissions: State where more GHGs are removed from the atmosphere than are emitted into it.Net zero CO_2_ emissions (also known as carbon neutral or carbon zero): State where anthropogenic carbon dioxide (CO_2_)
emissions are balanced globally by anthropogenic CO_2_ removals over a specified period.Net zero emissions: State where anthropogenic emissions of GHGs to the atmosphere are balanced by anthropogenic removals over a specified period.

### 1.2. Evolution of Corporate Climate Change Reporting

Following a major oil spill off the coast of California in 1969, there was pressure for companies’ corporate social responsibilities (CSR) to broaden beyond social impact, to include environmental issues. During the 1980s and 1990s, a series of supranational events representing efforts to set international climate-related standards, including the creation of the Intergovernmental Panel on Climate Change (IPCC—a UN body developed to assess the science related to climate change and develop recommendations for GHG emission reduction), the United Nations (UN) adoption of the Montreal Protocol, the UN Framework Convention on Climate Change, and the Kyoto Protocol, amongst others, resulted in a rising level of awareness of environmental issues on an international scale, and indirectly contributed to the institutionalization of CSR with an environmental focus (e.g., in 1992 the Business for Social Responsibility association was founded which included a focus on the preservation and restoration of natural resources) [6]. The 2000s saw the creation of the UN Global Compact initiative (which guides companies to align their strategies with environmental principles) and the adoption of the Millennium Development Goals, while 2015 saw the replacement of these with the Sustainable Development Goals (SDGs) and the signing of the Paris Agreement, an international treaty with the goal to limit the rise in mean global temperatures to below 2 °C [8,9]. 

Addressing climate change impact as part of CSR became a strategic consideration for companies in view of growing international, political, and public pressure. In some regions, elements of climate-related disclosures also became legislation (e.g., Directive 2014/95/EU of the European Parliament requiring large companies to disclose non-financial information; the UK Companies Act 2006 Strategic Report and Directors’ Report Regulations 2013; US Securities and Exchange Commission requiring some climate change disclosures) [10,11]. In response to companies’ obligation to quantify their commitments to CSR, the term Environmental, Social, and Governance (ESG) reporting began to gain traction, focusing on setting targets, measuring impacts, and reaching certain performance metrics. In particular, concerns about climate change prompted the inclusion of GHG emission reporting in company ESG reports [12].

In addition to the forementioned international treaties, a number of standards have been developed to aid companies in ESG reporting, including:Global Reporting Initiative (GRI) standards: International standard that aids businesses in reporting on issues including climate change, human rights, and corruption (1997) [13];Sustainability Accounting Standards Board (SASB): Industry-specific disclosure standards across ESG topics with a focus on sustainability and financial fundamentals (2011) [14];Taskforce for Climate-related Financial Disclosure (TCFD): Provides companies with recommendations for reporting climate-related financial risks (2015) [15].

These standards provide companies with a framework to report their environmental and social impact. However, there was a need to develop a standardized accounting method for measuring and reporting corporate climate change impact in the form of GHG emissions. Collaboration between the World Resource Institute and World Business Council for Sustainable Development resulted in the launch of the GHG Protocol in 1998 [16]. The protocol divides emissions into three scopes: Scope 1—direct emissions from owned or controlled sources (e.g., fuel combustion, company vehicles); Scope 2—indirect emissions from the generation of purchased energy; and Scope 3—indirect emissions that occur along the supply chain (consists of fifteen categories, e.g., transportation, distribution, and waste management). Emissions are usually reported in carbon dioxide equivalents (CO_2_e), which converts other GHG gases to the equivalent amount of carbon dioxide with the same climate change potential.

To aid companies to set targets and report their emissions, several voluntary non-profit initiatives have also gained prominence. For example, the Science Based Targets initiative (SBTi) is an international collaboration to help companies develop emission reduction targets in line with climate sciences [17]. The Carbon Disclosure Project (CDP) is a non-profit organization that allows companies to report on their environmental (e.g., climate change) impact, and then awards companies a score from A to F, with the aim of motivating positive change [18].

### 1.3. Literature Review: Pharmaceutical Industry GHG Emissions

A review of the current literature on GHG emissions in the pharmaceutical industry shows that it is limited and varies in conclusions. Previous studies of pharmaceutical company GHG emissions reported wide variations in emissions amongst global companies in 2012–2015 [19], and improvements in emissions while maintaining revenue, in 2017–2018 [20]. Two studies published on the Chinese pharmaceutical industry noted that energy-related CO_2_ emissions had increased by 140% between 2000 and 2016 [21], and that the potential emissions reduction potential of the Chinese pharmaceutical industry is larger than the emissions for the entire UK [22].

The literature on pharmaceutical company CSR/ESG reporting on GHG emissions indicates a limited and varied body of research. Some studies paint a picture of a lack of completeness and transparency of reporting as well as an inconsistency in reporting standards used [23,24]. Research on the environmental disclosures of the Indian pharmaceutical sector in 2014 found that only two of the nine pharmaceutical companies included in the study disclosed their emissions [25]. However, CSR reporting is evolving rapidly, and some analyses indicate that the pharmaceutical industry outperforms other industries in terms of their reporting and disclosures [26]. Indeed, three pharmaceutical companies were included in the EcoAct 2020 list of top companies for climate disclosure [27]. A survey of environmental reporting and internal sustainability policy of 17 pharmaceutical companies noted that companies are making progress, specifically in the area of green chemistry and technology [28].

The current literature on pharmaceutical company GHG emissions is limited; however, studies to date hint at the significant climate change contribution of the industry. From our review of the literature, there are no existing studies that examine the climate change targets that pharmaceutical companies are setting, the strategies they are developing and/or implementing to reach these targets, or how their emissions are changing over time. In light of health systems’ net zero targets regarding emission reductions along their supply chain, there is a need for up-to-date, peer-reviewed analyses of pharmaceutical companies’ CSR reporting, especially related to future climate change targets, GHG emission reporting, and the strategies they are using to reduce emissions. Such an analysis is important in understanding the nuances of reporting and reducing GHG emissions in the pharmaceutical industry that cannot be generalized from literature on other sectors.

### 1.4. Research Questions

To address this gap, we ask:(a)What public targets have pharmaceutical companies set to reduce their GHG emissions and are these aligned with international standards?(b)What GHG emissions are pharmaceutical companies reporting, with what (if any) reductions, and using which reporting standards?(c)What steps are pharmaceutical companies taking to reduce their GHG emissions?

To our knowledge this is the first study to assess the climate change targets and emission reduction strategies of leading pharmaceutical companies. We found that pharmaceutical companies are setting climate change targets and reducing emissions via a range of strategies. Reporting of emissions varies, with scope to track accountability to targets, improve consistency of reporting, and collaborate on novel solutions.

## 2. Materials and Methods

We used a content analysis approach to guide data collection and analysis. Content analysis is a “a research method that provides a systematic and objective means to make valid inferences from verbal, visual, or written data in order to describe and quantify specific phenomena” [29] (p. 314). The steps involved in content analysis, namely, sampling, data collection, and analysis, are described below [30].

### 2.1. Sample

We sought a purposive sample of pharmaceutical companies aiming to encompass the majority of revenue in this sector and a range of size, location, and product focus. To do this we identified the top 20 pharmaceutical companies by 2020 revenue [31]. This sample covers a significant proportion of the revenue, and therefore activity, in this industry [32].

### 2.2. Data Collection

Our main source of data was publicly available self-reported company documents published in 2020 or 2021. Between November and December 2021, one author (AB) visited the global website for each of the included pharmaceutical companies, and navigated to and downloaded web pages and documents likely to report on environmental considerations. This was supplemented with publicly available information on greenhouse gas (GHG) emissions and their baseline values from company climate change reports on the website of the non-profit Carbon Disclosure Project (CDP). No primary data were collected as this was outside the scope of this research.

AB and AJ then read those documents and extracted data manually, using a data extraction form (summarized in Appendix A), on basic company information, climate change targets and the standards they were aligned with, GHG emissions (reported value for the latest 12-month period, baseline value and year of reporting), frameworks and standards used to report emissions, and any initiatives or strategies companies reported implementing to reduce emissions. To ensure nothing was missed, the search function was used in the relevant application (e.g., Adobe Acrobat) to search for the following terms: “environmental sustainability”, “GHG emissions”, “carbon footprint”, “carbon neutral”, “net zero”, “energy”, “scope 1”, “scope 2”, and “scope 3”. This provided a check on manual extraction, and additionally allowed us to identify specific instances where companies mentioned their targets, emissions, or strategies (e.g., where a company reported their GHG emissions in the text rather than in a table of environmental key performance indicators).

Given the focus of this study on GHG emissions, other environmental reporting (e.g., water use, waste production), or targets to improve other environment indicators, were excluded.

### 2.3. Analysis and Synthesis

We used content analysis to analyze and interpret data extracted from documents. This approach has previously been used successfully to assess corporate environmental disclosures [33]. Our focus was on providing a descriptive summary of company climate change targets, the emissions they are reporting (according to which standards), and strategies they reported to reduce these emissions, as well as descriptive statistics reflecting the proportion of companies engaging in the reporting of these issues. We also determined the percentage change (either increased or decreased) in reported company emissions from the baseline year of reporting.

Public targets to reduce GHG emissions were documented (with target date) and categorized as net zero (those companies that made statements including “net zero”, “climate positive”, “reduction in GHG emissions to zero”), carbon neutrality (those companies that made statements including “carbon neutral”, “carbon zero”, “zero carbon emissions”) or other reduction (e.g., percentage emission reduction). We documented whether these targets were aligned with international treaties, policies, or initiatives.

Scope 1, 2, and 3 GHG emission values reported by the companies for the most recent year and respective baseline years were documented (Appendix A). When companies reported both location-based and market-based scope 2 emissions, the location-based value was included in the analysis because it is similar to the method required for scope 2 reporting in the original GHG Protocol Corporate Standard and reflects “the average emission intensity of grids on which energy consumption occurs” [34] (p. 4). The change in the reported emissions of each company was calculated as a percentage increase or decrease from the reported baseline year for each company. Proportions of companies that reported on scope 1, 2, and 3 emissions as well as the proportion that documented an improvement in the metric were determined by dividing by the total number of companies included in the study.

Standards or tools that companies used to report emissions were documented and the completeness with which these were applied was assessed.

Finally, we extracted information on strategies that companies reported implementing to reduce their emissions. These were grouped according to common themes and the GHG Protocol scope targeted by the strategy.

## 3. Results

### 3.1. Overview of Pharmaceutical Companies

Nine of the companies included in the study had head offices in the USA, eight in Europe, two in Japan, and one in Israel (Table 1). Across the 20 companies, annual revenue ranged from USD 11.0 billion to USD 47.5 billion and employee numbers from >11,000 to >130,000. Eighteen companies produce branded medication and two produce generics (Viatris and Teva); one company (Boehringer Ingelheim) is privately owned, and the other nineteen are publicly listed, one of which (Novo Nordisk) is controlled by a foundation focused on sustainability, life sciences, and health.

### 3.2. Location of Data

All publicly listed companies are legally required to release an annual report on a yearly basis to communicate their financial statements, governance, and other aspects to shareholders. Some companies have taken to releasing integrated reports which include reports on finance, Corporate Social Responsibility (CSR), and Environmental, Social, Governance (ESG) aspects. Others produce separate reports or policy documents/position statements on these indicators. We identified a total of 86 documents with data on greenhouse gas (GHG) emissions from company websites, including 14 annual reports, 15 Environmental, Social Governance (ESG), Corporate Social Responsibility (CSR), or sustainability reports, 2 integrated reports, 7 policy documents, 28 position statements, 6 data summary reports, and 7 infographics, and selected additional information obtained directly from company websites (for example, Boehringer Ingelheim included information on GHG emission reporting and commitments on their website’s sustainability tab and not in any published reports).

### 3.3. Reported GHG Emission Reduction Targets

Although outside the scope of this research, it must be noted that seven companies reported progress on completed GHG emission reduction target periods that culminated in 2020 or earlier.

Nineteen companies reported current targets to reduce their GHG emissions through commitments to one or more of carbon neutrality, net zero, or percentage emission reduction. Ten companies had made commitments to carbon neutrality and eight to net zero emissions between 2025 and 2050 (Appendix A). Companies set different targets for different scopes; for example, they might commit to net zero in scope 1 and 2, and carbon neutral in scope 3. Companies reported their carbon neutrality and net zero targets in the form of differing statements, for example, “carbon neutral across scope 1 and 2”, “net zero impact on climate”, or “carbon zero without using offsets” (Appendix A).

Sixteen companies pledged to reduce GHG emissions by a certain percentage by a range of target years (Figure 1). In some cases, these percentage emission reductions reflect the percentage of emissions companies need to reduce to reach their carbon neutral or net zero targets. In other cases, companies made percentage emission reduction pledges instead of pledges to carbon neutrality or net zero. Five companies committed to 100% GHG emission reductions in scope 1 and 2, one of which additionally committed to a 100% reduction in selected scope 3 emissions, namely, transport and distribution. One company (Gilead Sciences) set a scope 3 percentage GHG emission reduction target, but did not report on scope 3 emissions (Appendix A).

### 3.4. Alignment of Company Targets with International Standards

The companies aligned their reporting and targets with the following treaties, policies, or initiatives (Appendix A), all of which aim to keep the rise in mean global temperature to below 2 °C, and preferably limited to 1.5 °C:20 reported aligning their climate change reporting with the United Nations (UN) Sustainable Development Goals;19 with the UN Global Compact initiative;18 with the Paris Agreement;16 with the Science Based Targets Initiative (SBTi);10 with the targets outlined in the Intergovernmental Panel on Climate Change (IPCC) Report;one with the 2 °C scenario;one with the commitments of the European Green Deal which aims to reduce emissions by 50% by 2030 and become climate neutral by 2050.

### 3.5. Progress on Targets: Scope 1, 2, and 3 GHG Emission Reporting Standards

All 20 companies stipulated that they measured and reported their emissions using The Greenhouse Gas (GHG) Protocol Corporate Accounting and Reporting Standard [16]. However, there were inconsistencies with the way in which this protocol was applied to scope 3. For example, while eleven companies included all 15 GHG Protocol categories for scope 3, three only included business travel or flights in their scope 3 reporting, and six companies did not report on scope 3 emissions at all.

Other international reporting standards, tools, and platforms that companies used included: the Carbon Disclosure Project (CDP; *n* = 18), the Sustainability Accounting Standards Board (SASB; *n* = 18), the Task Force on Climate-Related Financial Disclosures (TCFD; *n* = 18), and the Global Reporting Initiative (GRI; *n* = 16). Some companies also reported using national standards such as the US Environmental Protection Agency GHG Reporting Program or reported having their own internal reporting standards. All companies reported using third-party auditors, commonly Environmental Resources Management Certification Verification Service (ERM CVS; *n* = 7), Klynveld Peat Marwick Goerdeler (KPMG; *n* = 2), PriceWaterhouseCoopers (PWC; *n* = 2), and Det Norske Veritas (DNV; *n* = 2).

### 3.6. Progress on Targets: Scope 1, 2, and 3 GHG Emissions and Reported Change from Baseline

All 20 companies reported on scope 1 and 2 emissions and fourteen on scope 3 emissions (Figure 2, data available in Appendix A).

Reductions in scope 1 emissions over a median of 5 years from the company’s baseline year of reporting were noted in 17 of the 20 companies. Six companies managed to reduce their scope 1 emissions by more than 20% from their respective baseline year of reporting. Nineteen companies reduced their location-based scope 2 emissions, with eight companies reducing emissions by 30% or more from their respective baseline years.

Fourteen companies reported on a range of scope 3 emission categories (see Appendix A). Nine of these fourteen noted a reduction from their respective baseline year.

### 3.7. GHG Emission Reduction Strategies

All 20 companies described varied emission reduction strategies that they had either implemented or planned to implement to reduce their emissions; see Table 2. The majority (18–20) of companies are implementing measures to reduce their emissions from energy use, and tackle their scope 1 and 2 emissions. There was a wider range of initiatives to reduce scope 3 emissions, of which companies engaged most consistently with optimizing waste and water management.

## 4. Discussion

This content analysis of public reports from 20 pharmaceutical companies demonstrates that the vast majority (nineteen companies, 95%) have published targets to reduce their greenhouse gas (GHG) emissions, aligned with a variety of international standards: 10 (50%) have committed to carbon neutrality and 8 (40%) to net zero emissions. All 20 companies are reporting on scope 1 and 2 GHG emissions using the widely used GHG Protocol Corporate Accounting and Reporting Standard, through a variety of corporate channels and the Carbon Disclosure Project (CDP). However, only 11 companies (55%) are comprehensively reporting on scope 3 (supply chain) emissions that typically make up the bulk of emissions. The majority of companies reported reductions in scope 1 (17/20; 85%), scope 2 (19/20; 95%), and scope 3 (9/14; 64%) emissions. They aim to meet their targets and reduce their emissions by implementing a range of emission reduction strategies, the most common being using renewable energy, improving energy efficiency in operations, sourcing raw materials sustainably, reducing and recycling waste, and reducing water use.

It is encouraging that all companies included in our study are reporting on their GHG emissions. However, reporting is inconsistent and missing data, especially relating to scope 3 emissions, in some of the companies. Our findings of incomplete reporting of GHG emissions by pharmaceutical companies are comparable with studies across other industries. For example, an analysis of food and beverage companies demonstrated that 7 of the largest 50 companies did not report emissions, and that reporting, especially of scope 3 emissions, is incomplete [35]. A case study of the tech industry found that corporate reports omit half of company GHG emissions [36], while a study of manufacturing companies found low levels of emission disclosure [37]. In addition, while all but one company in our study had reported climate change targets, few of these targets covered scope 3 emissions, again comparable with other industry targets [35,38].

The inconsistent reporting of scope 3 emissions (acknowledging the challenges associated with obtaining supplier data and accurately calculating them) can lead to erroneous comparisons and conclusions about the relative contributions of industries to climate change. For example, previous claims that the pharmaceutical industry is 55% more emission intensive than the automotive industry have been contested on the basis that scope 3 emissions of the two industries were not included in these calculations and comparisons [19,39]. There are a range of voluntary sustainability reporting standards available that are incorporated into the reporting models of companies included in our sample. For the most part, companies engaged with a combination of Global Reporting Initiative (GRI) standards, Sustainability Accounting Standards Board (SASB), and the Task Force on Climate-Related Financial Disclosures (TCFD). Using sustainability standards has been shown to improve sustainability disclosures, and standards have also been shown to compliment, rather than substitute, each other [40]. For example, while GRI supports broad disclosures on organizational impacts, SASB focuses on financially material issues [41]. These standards are internationally recognized; however, we note that some companies also used national reporting mechanisms. The choice of reporting mechanisms (i.e., international versus national; single standard versus a combination) is usually defined by a company’s ESG narrative, their audience, location, industry trends, and resources available to produce such reports. In an effort to harmonize these multiple reporting standards, the International Sustainability Standards Board (ISSB) is currently developing a single guiding reporting framework [42].Third-party assurance is also important to give credibility to company’s reported data; all companies reported that their data had undergone auditing.

For the most part, companies showed favorable scope 1 and 2 emission reductions from their respective baseline years of reporting. These reductions indicate that reaching company climate change targets (especially for scope 1 and 2 emissions) is feasible, although reductions may need to be expedited if companies are to reach the range (20–100%-see Figure 1) of commitments to percentage emission reduction in the next 3–13 years as publicized. These findings are compatible with a 2021 study that demonstrated that pharmaceutical companies have been able to reduce their own emissions while maintaining revenue earned [20], as well as a study that demonstrated the large emission reduction potential of the Chinese pharmaceutical industry [22]. The demonstration of reduction in emissions by companies in our sample, however, contrasts with a 2015 study that demonstrated a disconnect between corporate carbon management practices and reduction in reported emissions, although this might point to recent advancements in strategies to reduce emissions and the delay between implementing reduction strategies and seeing an impact on emissions [43]. Controlling the emissions in a company’s own operations is relatively straightforward, as reflected in widespread improvements accomplished by efficiency gains in operations, use of renewable energy, and switching to an electric vehicle fleet.

Scope 3 emissions were the largest share of emissions in this study, despite the varied, and in some cases, incomplete, reporting by companies, with variable improvements in emission reduction. It is, therefore, encouraging to see that many of the strategies proposed by companies to reduce emissions, target scope 3. Some of the proposed solutions to reducing these emissions reported by the Science Based Targets initiative (SBTi) include supplier engagement (in line with climate science), procurement of products from suppliers with a lower carbon footprint, designing products that are more efficient and integrating circular economy principles to reduce the lifecycle emission intensity of the product, embedding sustainability into procurement policies, customer engagement, and assigning a monetary value to carbon by using an internal carbon price to provide a financial incentive for low-carbon business models [44]. Many of the pharmaceutical companies studied are engaging with these principles and other solutions unique to the pharmaceutical industry and drug manufacturing process. The emission reduction potential of green chemistry principles, streamlined batch production, and optimizing pharmaceutical packaging have been demonstrated through several life cycle assessments published in the literature [45,46,47].

Many of the pharmaceutical companies reported engaging in carbon offsetting measures as a means of mitigating their emissions. Carbon offsetting is not without criticism. For example, tree planting is a popular method of carbon offsetting, but concerns have been raised that it promotes monoculture, reduces natural diversity, and, in some cases, displaces indigenous people from their land [48]. While offsetting may be necessary in some cases, it is recommended by the GHG Management Hierarchy as a last resort and only after every measure possible has been undertaken to reduce the company’s own emissions [49].

Much of the drive to reduce emissions at the industry level comes from external drivers, including corporate image and a rise in legislation mandating reporting, as well as internal drivers, such as sustainability-orientated leadership and management, green human resource management, cost savings, and innovative research and development processes [25,50]. Indeed, a literature review of CSR performance in the pharmaceutical industry demonstrated the role of strict legislation in improving the environmental reporting of companies [51].

### 4.1. Strengths and Limitations

To the best of our knowledge, this is the first study to examine the climate change targets and emission reduction strategies of leading pharmaceutical companies, which we assessed using content analysis. This is also the first study to analyze one of the major health system suppliers in this context. We add to the body of literature on GHG emission reporting.

We focused on the top 20 companies in terms of revenue. This sample covered a significant proportion of industry revenue, with the top 25 companies accounting for 73% of all pharmaceutical sales in 2015 [32]. This provides a diverse sample, but does not represent the entirety of the industry. In particular, our sample did not include companies whose headquarters are located in developing economies or smaller pharmaceutical companies, which may struggle to undertake the activities and reporting undertaken by larger pharmaceutical companies [52].

The study was reliant on secondary data from publicly available self-reported company and CDP reports. While there are limitations in relying on self-reported company data, we note that the global emission disclosure system is based on self-reporting using international standards and audited by independent third parties, and these data are, therefore, suitable for our analysis, which aimed to describe what targets, emissions, and strategies companies are reporting.

The location of where targets/emissions are reported by companies varied. The number and range of reports, policies, position statements, data summaries, and infographics that companies employed to report on emissions can make it challenging to fully appreciate the extent and completeness of reporting. We combined manual and computer-generated data extraction and double-checked our data collection; however, it is possible that we missed data.

Company data and reduction initiatives are constantly changing, and annual reporting only captures a snapshot of the action that is being undertaken. Given that we used 2020/2021 reports, the data (especially of emission values) will change as soon as updated reports are published; however, we believe that the conclusions drawn on how emissions are reported and types of emission reduction strategies will be of use regardless of the actual data values.

It is also possible that the information in company reports does not capture all the company’s actions towards environmental sustainability. Conversely, companies may give preference to initiatives or data that present their emission reduction activities in a favorable light or communicate positively about environmental performance when actual performance is poor (“greenwashing”) [53]. It is important to consider that what a company states in its reports is often a reflection of what it aspires to achieve, rather than how it might actually be operating. For example, a company may state that it aims to increase use of renewable energy, but has yet to do so. We therefore set out to separate aspiration from progress in our research.

We did not make direct comparisons between companies’ GHG emissions (whether absolute values or reductions) due to the different reported baseline years, differing definitions of emissions, different baseline emissions and resource use, differing operational structure across companies, and different size and scope of companies’ production and employee numbers, all of which affect their reported values, as well as a lack of consistent use of a standard reporting framework. For example, companies that already have low baseline emissions (e.g., because they have already completed a climate change target period, as seven companies in our study had) would find it more difficult to further reduce their emissions and thus their targeted reduction in emissions would be lower.

It was not possible to distinguish reported information on emissions from all the divisions within one company (i.e., companies that produce pharmaceuticals as well as other products do not report breakdowns by division). We also did not provide comparisons between the GHG emissions of pharmaceutical companies and companies in other commercial sectors. As with the challenges noted above of comparing companies within the same sector, companies in different sectors also run different operations, are located in different geographic locations, report their emissions with varying completeness, and have different baseline emissions. Thus, making comparisons of progress in different sectors has limited value.

In addition to the differences in reporting company GHG emissions, we note the intrinsic limitations of measuring GHG emissions, even when using standards such as the GHG Protocol. These include definitions of organizational boundaries to include in the reporting, limitations in the conversion factors used, the availability of robust data, and the constantly changing nature of a company’s operations and supply base. We therefore acknowledge that any organization’s reported GHG emissions will never be absolutely accurate, but are, nevertheless, a best estimate within the bounds of the limitations of any modeling system.

Finally, the 2020/2021 public reports include data gathered during the COVID-19 pandemic. The accompanying restrictions, in particular lockdowns, as well as the increase in home-working likely led to a decrease in reported emissions (in particular related to scope 3, which includes business travel) compared to the baseline years. This decrease may not be sustained in subsequent years as COVID regulations are eased.

### 4.2. Implications for Practice

The targets that companies are reporting setting to reduce their emissions are varied. It would be beneficial if stakeholders, including academics, policymakers, and industry, could collaborate to agree on how to operationalize climate change targets (and the terms used to refer to these targets) and what each involves. Climate change targets need to be more thorough, including the type of emissions, timing, rate of emission reduction, specific reductions envisioned, and expected outcomes. All companies need to include comprehensive scope 3 emissions in their climate change targets. These targets must align with international standards and be developed using climate science, and regulators must hold companies accountable to these targets. Independent auditing of companies’ progress towards their emission reduction targets, such as through the CDP platform, is important as it promotes accountability and validation of their progress.

It is encouraging to see that all companies in our sample use a single reporting standard to measure and report their emissions, namely the GHG Protocol. While the use of a single standard is a first step towards uniformity in reporting, the way companies apply the protocol differs, especially when reporting scope 3 emissions. This can lead to incomplete values and reports of success, and needs to be standardized across the industry. The wide array of reporting standards available (SASB, TCFD, GRI etc.,) gives companies options on how they report and companies should be clear on why they are using particular standards and how these aid reporting. A single, harmonized, international reporting standard (as the ISSB is developing) would undoubtedly simplify company reporting, both for the company and for readers of the reports.

Companies are implementing a range of strategies to reduce their emissions. We recommend that companies engage with industry bodies, to share these ideas, and spread and scale-up the solutions across the industry. Given that scope 3 emissions form the bulk of companies’ emissions, we recommend that companies engage with their suppliers to improve supplier reporting and emission reduction. Small to medium-sized pharmaceutical companies may need support from governments and policymakers given their potentially limited capacity to report on and reduce their environmental impact.

### 4.3. Implications for Further Research

This study was largely descriptive in nature, and is intended to draw attention to how companies are reporting and reducing emissions. Further research is needed to enhance and extend this analysis. This could be through useful engagement with relevant theory (e.g., relating to greenwashing [53]), and/or use qualitative (e.g., supplementing company data with interviews of representatives from pharmaceutical companies) and quantitative (e.g., performing econometric comparisons of driving factors of emission reduction strategies or analyzing differences by country and regions) methods. Our study focuses on one time period. It would be valuable to repeat a similar study on emissions of pharmaceutical companies in the future to see the extent to which they were able to reach their targets, and whether they have gone on to set more stringent goals. While we would be cautious with making comparisons between companies, it would be possible to make intracompany comparisons over the next few years by adopting this approach.

An analysis of the policy and regulations surrounding emission reporting of industries, how these are being implemented and acted upon, and where incentives might play a role, for example, tax rebates or priority on tenders for companies that embody environmental sustainability, could provide interesting insights.

More work is needed to refine the methodology for supply chain emission reporting (scope 3) along with more engagement between academics, policymakers, and the industry with ways in which these emissions can be reduced (and the effectiveness of different interventions). We recommend exploring company climate change targets and reported GHG emissions in other industries, especially other health care suppliers (e.g., medical device companies).

## 5. Conclusions

Pharmaceutical companies contribute significantly to emissions related to the health sector. This urgently needs addressing. The top 20 companies in terms of annual revenue are, to varying degrees, working towards reducing their emissions through a range of publicized commitments to carbon neutrality, net zero, and percentage emission reduction, and by implementing a range of emission reduction strategies. These targets are aligned with international standards. However, there remains significant scope for improvement and further action, especially in scope 3 emission reporting and reduction. Time is short. We urge pharmaceutical companies to extend and act on their targets so that we can, in line with the Paris Agreement, limit mean global temperature rise to well below 2 °C.

## Figures and Tables

**Figure 1 ijerph-20-03206-f001:**
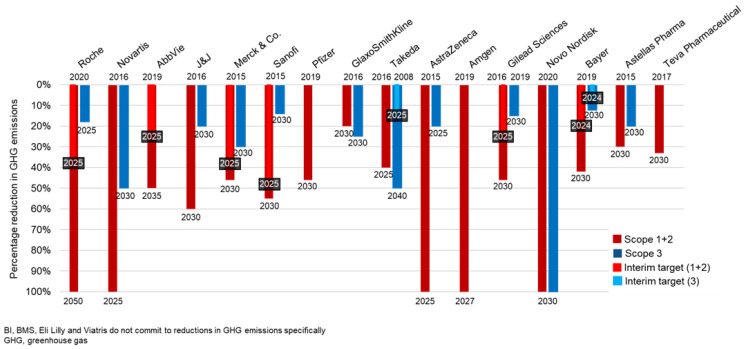
Commitments to percentage greenhouse gas (GHG) emission reduction in scope 1, 2, and 3 emissions by a range of target years by 16 pharmaceutical companies that made them.

**Figure 2 ijerph-20-03206-f002:**
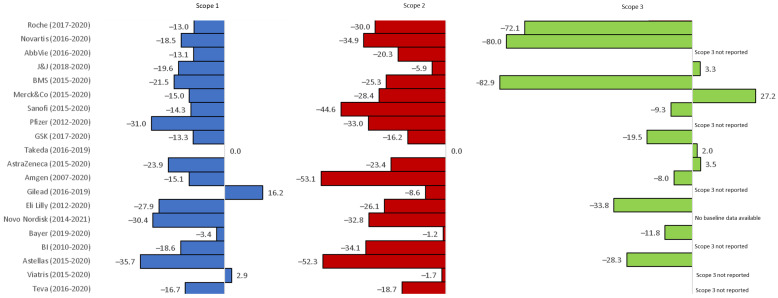
Percentage change over reporting period (in brackets) in company scope 1, 2, and 3 GHG emissions (Roche, Novartis, and BMS only reported business travel/flights in scope 3 emissions).

**Table 1 ijerph-20-03206-t001:** Overview of the 20 pharmaceutical companies included in the study.

Company Name	Headquarters	2020 Annual Revenue (Billion USD)	Number of Employees
Roche	Switzerland	47.5	>90,000
Novartis	Switzerland	47.2	>100,000
AbbVie	USA	44.3	>47,000
Johnson & Johnson (J&J)	USA	43.1	>130,000
Bristol Myers Squibb (BMS)	USA	41.9	>30,000
Merck & Co.	USA	41.4	>74,000
Sanofi	France	35.8	>100,000
Pfizer	USA	35.6	>78,000
GlaxoSmithKline (GSK)	UK	30.6	>99,000
Takeda	Japan	27.9	>50,000
AstraZeneca	UK	25.5	>76,000
Amgen	USA	24.1	>22,000
Gilead	USA	23.8	>11,000
Eli Lilly	USA	22.6	>35,000
Novo Nordisk (NN)	Denmark	19.4	>45,000
Bayer	Germany	18.9	>99,500
Boehringer Ingelheim (BI)	Germany	16.5	>47,000
Astellas	Japan	11.5	>15,000
Viatris	USA	11.5	>45,000
Teva	Israel	11.0	>40,000
Total	20	580.4	>1,200,000

**Table 2 ijerph-20-03206-t002:** Emission reduction strategies reported by all 20 pharmaceutical companies to reduce scope 1, 2, and 3 greenhouse gas emissions.

Greenhouse Gas Emission Scopes and Categories		Emission Reduction Strategy *	Company	Number (Proportion)
Scopes 1 and 2	Energy	Increase renewable energy purchased/install onsite renewable energy	Roche; Novartis; AbbVie; J&J; BMS; Merck & Co; Sanofi; Pfizer; GSK; Takeda; AZ; Amgen; Gilead; Eli Lilly; Novo Nordisk; Bayer; BI; Astellas; Viatris; Teva	20 (100%)
		Reduce operational energy use, e.g., use of energy efficient equipment, timers on equipment	Roche; Novartis; AbbVie; BMS; Merck & Co; Sanofi; Pfizer; GSK; Takeda; AZ; Amgen; Gilead; Eli Lilly; Novo Nordis; Bayer; BI; Astellas; Viatris; Teva	19 (95%)
		Optimize energy use in building design e.g., optimize heating, ventilation and air conditioning, install LED lights, obtain ISO/LEED certification	Novartis; AbbVie; J&J; BMS; Merck & Co; Pfizer; Sanofi; Takeda; AZ; Amgen; Gilead; Eli Lilly; Novo Nordisk; Bayer; BI; Astellas; Viatris; Teva	18 (90%)
		Optimize own manufacturing process through green chemistry principles, e.g., increased efficiency of process, reduced water and energy use	Roche; Novartis; AbbVie; J&J; BMS; Merck & Co; Sanofi; Pfizer; GSK; Takeda; AZ; Amgen; Gilead; Eli Lilly; Novo Nordisk; Bayer; BI; Astellas; Teva	19 (95%)
	Company-owned vehicles	Convert to energy efficient vehicle fleet, e.g., hybrid or electric	Novartis; AbbVie; BMS; Merck & Co; Sanofi; GSK; Takeda; AZ; Amgen; Gilead; Eli Lilly; Novo Nordisk; Astellas	13 (65%)
Scope 3	Purchased goods/services	Implement sustainability criteria in vendor selection processes and encouraging suppliers to disclose environmental performance (e.g., through platforms such as EcoVadis)	Roche; Novartis; AbbVie; J&J; BMS; Merck & Co; Sanofi; Pfizer; GSK; Takeda; AZ; Amgen; Gilead; Eli Lilly; Novo Nordisk; Bayer; BI; Astellas; Viatris; Teva	20 (100%)
		Assist suppliers to convert to renewable energy in their manufacturing processes	Roche; Novartis; J&J; Merck & Co; Sanofi; Pfizer; GSK; AZ; Amgen; Novo Nordisk; Bayer; Viatris; Takeda	13 (65%)
		Source raw materials responsibly (e.g., reduce hazardous substances and precious metals in production processes, use recycled materials in packaging, and ensure protection of biodiversity in sourcing)	Roche; Novartis; AbbVie; J&J; BMS; Merck & Co; Sanofi; Pfizer; GSK; Takeda; AZ; Amgen; Gilead; Eli Lilly; Novo Nordisk; Bayer; BI; Astellas; Viatris; Teva	20 (100%)
		Reduce water consumption or engage in water recycling or reuse projects	Roche; Novartis; AbbVie; J&J; BMS; Merck & Co; Sanofi; Pfizer; GSK; Takeda; AZ; Amgen; Gilead; Eli Lilly; Novo Nordisk; Bayer; BI; Astellas; Viatris; Teva	20 (100%)
	Capital goods	Bought buildings are assessed for environmental sustainability and energy efficiency	Roche; AbbVie; BMS; Sanofi; Pfizer; AZ; Amgen; Gilead; Merck & Co	9 (45%)
	Transportation and distribution	Converting from air to sea/land distribution	Novartis; BMS; Merck & Co; Sanofi; Takeda; AZ; Eli Lilly; Novo Nordisk; Bayer	9 (45%)
		Implement technology to determine lowest carbon route of transport and distribution	J&J; Eli Lilly	2 (10%)
	Waste production	Efforts to reduce waste generated from manufacturing process	Roche; AbbVie; Novartis; J&J; BMS; Merck & Co; Sanofi; GSK; Pfizer; Takeda; AZ; Amgen; Gilead; Eli Lilly; Novo Nordisk; Bayer; BI; Astellas; Viatris; Teva	20 (100%)
		Improvement of recycling rates and reuse of waste	Roche; Novartis; AbbVie; J&J; BMS; Merck & Co; Sanofi; Pfizer; GSK; Takeda; AZ; Amgen; Gilead; Eli Lilly; Novo Nordisk; Bayer; BI; Astellas; Viatris; Teva	20 (100%)
		Reduce waste to landfill	Roche; AbbVie; J&J; BMS; Merck & Co; Sanofi; GSK; Takeda; AZ; Amgen; Gilead; Astellas; Eli Lilly; Novo Nordisk; BI; Viatris	16 (80%)
		Implement E-labelling to reduce packaging waste	BMS; Takeda; AZ; Astellas	4 (20%)
		Efforts to reduce food waste	Novartis; J&J; BMS; Sanofi; Gilead	5 (25%)
		Implement auditing of waste and recycling vendors	Roche; AbbVie; Pfizer; Astellas	4 (20%)
	Business travel	Reduce business travel	Roche; Pfizer; BMS; Sanofi; GSK; Takeda; AZ; Novo Nordisk; Bayer; BI	10 (50%)
		Encourage train instead of air travel	Sanofi; Merck & Co	2 (10%)
	Employee commuting	Encourage remote working	Novartis; BMS; Merck & Co; Sanofi; GSK; Viatris;	6 (30%)
		Provide alternative low-carbon ways to get to work (e.g., electric buses to transport staff, charging points at work)	Novartis; Sanofi; Takeda; Bayer	4 (20%)
	Leased assets	Assess leased facilities for energy efficiency	Merck & Co	1 (5%)
	Processing and use of sold products	Use propellant with lower climate impact in inhalers	GSK; AZ	2 (10%)
		Encourage recycling of insulin syringes	Novo Nordisk	1 (5%)
	End-of-life management	Introduce take-back programs for products/promote responsible consumer disposal	Roche; J&J; Sanofi; Pfizer; AZ; Amgen; Gilead; Eli Lilly; Novo Nordisk	9 (45%)
		Monitoring system or wastewater treatment systems for pharmaceuticals in the environment	Roche; Novartis; AbbVie; J&J; BMS; Merck & Co; Sanofi; Pfizer; GSK; Takeda; AZ; Amgen; Gilead; Eli Lilly; Novo Nordisk; Bayer; BI; Astellas; Viatris; Teva	20 (100%)
All:	Carbon offsets	Support tree planting, anti-deforestation initiatives, water and waste management projects etc.,	Novartis; J&J; BMS; Merck & Co; Pfizer; GSK; Takeda; AZ; Amgen; Eli Lilly; Bayer; BI; Teva	13 (65%)

* Some strategies will likely reduce emissions across more than one scope, for example, optimizing the manufacturing process through green chemistry principles will aid in reducing scope 1 and 2 energy emissions by improving the energy efficiency of the process, as well as scope 3 emissions from raw material acquisition; however, for the sake of simplicity, they are only included in the table once.

## Data Availability

Data used in this study can be found in public pharmaceutical company reports. The data presented in this study are available in the article and Appendix A.

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
