# Peer review of "Pharmaceutical Company Targets and Strategies to Address Climate Change: Content Analysis of Public Reports from 20 Pharmaceutical Companies"

_ijerph, 2023, doi:10.3390/ijerph20043206_

Round 1

Reviewer 1 Report

This is an interesting and timely article which provides a contemporary analysis of the 20 largest pharmaceutical companies' climate change commitments. The article is well written and presented throughout. The following minor points could further improve the article prior to publication 

-         It would be helpful in the introduction to also provide a comparison of GHG contributions in other commercial sectors. It is unclear as to whether the pharmaceutical industry is under or out-performing these in terms of progress.

-         Additional methodological detail on the suitability for content analysis (lines 104-105 would be welcome) to assist those readers who are not familiar with this approach.

-         The accuracy of tools used to assess Scope 3 emissions is unclear and could be further expanded in the discussion section. This is important because of the variability in reporting and proportion of emissions share.

- Further detail on the respective role of national/international reporting mechanisms would be helpful in the discussion. 

Author Response

REVIEWER 1

This is an interesting and timely article which provides a contemporary analysis of the 20 largest pharmaceutical companies' climate change commitments. The article is well written and presented throughout.

We thank the reviewer for this positive feedback.

It would be helpful in the introduction to also provide a comparison of GHG contributions in other commercial sectors. It is unclear as to whether the pharmaceutical industry is under or out-performing these in terms of progress.

Our main focus was understanding climate change targets, GHG emission reporting, and strategies to reduce emissions in an important supplier of healthcare systems (i.e., pharmaceutical industry). Nevertheless, we have added some comparisons of GHG emissions reporting with those of other commercial sectors in the Discussion section. Alongside this we have also noted that it is difficult to make meaningful comparisons between sectors as they all run different operations, are located in different geographic locations, report their emissions with varying completeness, and have different baseline emissions.

Additional methodological detail on the suitability for content analysis (lines 104-105 would be welcome) to assist those readers who are not familiar with this approach.

Thank you for noting this. We have added an explanation of content analysis in the Methods section.

The accuracy of tools used to assess Scope 3 emissions is unclear and could be further expanded in the discussion section. This is important because of the variability in reporting and proportion of emissions share.

This is an important point - we agree. At present, the GHG Protocol is the main tool used to measure all GHG emissions, including Scope 3. We have added more information about this in the Introduction and in the limitations section of the Discussion. We have also indicated in Figure 2 when the data presented are from a subset of overall Scope 3 emissions e.g., only business travel. This data is also available in the Supplementary Tables.

Further detail on the respective role of national/international reporting mechanisms would be helpful in the discussion. 

Thank you for pointing this out. The array of voluntary national and international reporting mechanisms can be confusing, and as we found in our results, while all companies use the GHG Protocol to guide measuring of emissions, they make use of a mixture of different standards (e.g., GRI/SASB) and platforms (e.g., CDP) to report them. Usually, companies choose between the different options based on what their ESG narrative is, their audience, location, and available resources. At present, work is being done to prepare a single international reporting standard to simplify reporting. We now outline the range of standards proactively in the Introduction. We have also added more information on these in the Discussion.

Author Response

REVIEWER 2

The literature review is entirely missing. There is maybe a specific reason, but the current version of the paper has no section reviewing any kind of academic literature. It means, consequently, that there is no definition of the terms, there is no justification of the literature gap, there is no evidence explaining how strong the research question is. In the literature review, authors should highlight arguments making the positioning of this research very clear.

In the current version, we have the feeling that the literature has not been entirely scrutinized. The very large field of CSR and climate change is not researched at all, although it includes a lot of substantial arguments to elaborate the research questions.

As a matter of examples, we provide three reasons why a literature review is needed (that list is not comprehensive) 1. A literature reviews helps to define terms. Here, there is no definition of “GHG emissions”, “net zero” net 108 zero”, “climate positive”, “reduction in GHG emissions” etc. All these terms must be seriously defined. 2. It clarifies the literature gap. Here the first research question mentions “international standards”, which requires a section about these standards in the literature review. This suggests some companies are following international standards, others are not. 3. It brings constructs to the research.

We thank the reviewer for this feedback. The reviewer rightly points out that there is very limited literature on the climate change impact of the pharmaceutical industry, and this gap is one that we set out to address through our research. Although primary manuscripts do not typically include a formal literature review, we have expanded the Introduction to outline key concepts, terminology, and standards likely to help the reader interpret our findings and make the literature gap clearer. We note that we are aware of no research that provides an overview of climate change targets and GHG emissions reported by pharmaceutical companies and the strategies they report using to tackle these emissions and reach their targets. Our article aims to provide a baseline on which additional research can be built and against which the reported progress of companies can be monitored. We have also extended the Discussion section with examples of how other industries are engaging with this area.

Research questions mention “GHG emission reduction”, which calls for a good understanding of what is driving GHG emissions (before discussing how to reduce the emissions). Thus, the literature review should report on drivers, processes, methods to measure GHG emissions across the different scopes 1/2/3, and provide constructs facilitating the comparison practice versus theory (because you cannot assess firms’ maturity in GHG emission without discussing the current knowledge about that particular field). These are only examples, justifying the need of adding a strong literature review section to that paper. This is indeed the biggest weakness of that paper.

As mentioned above, we have added sections to the Introduction on terminology, the evolution of CSR/ESG, methods to measure GHG emissions across the scopes, and current, limited literature on the pharmaceutical industry.

Research strategy is not clear and the methodology section does not contain the basic/necessary elements describing the research approach. The three common approaches to conducting research are quantitative, qualitative, and mixed methods. The researcher anticipates the type of data needed to respond to the research question (Williams, 2007).

Methodology requires major revisions, especially because the data collected come strictly from a collection of secondary sources. This is too weak to justify an adequate research validity. Indeed, concluding a research paper only based on firms legal declarations is not a robust approach, especially to explore firms’ actions on climate change. More precisely, considering annual reports can be a good way to better understand the context (in one word, it is necessary), but it is not sufficient because the nature of the annual reports makes them look like a marketing tool rather than a fair source of information. Green washing positions are spread extensively within annual reports (Netto et al, 2020). Quality of data, especially data coming from people working in these companies, could strengthen the results. For instance, interviews of key people in these firms could benefit to a better understanding of the real actions taken and add granularity in the way these actions relate to annual reports. That could complement the first stage of that research, based only on secondary data.

The analysis: it focuses only on the description of annual reports. It is therefore based on a simple description and not an analytical approach. The deepness of the analysis has to be strengthened, which would be feasible if more data were collected.

Thank you for these comments. This study uses content analysis (an accepted research approach [1-3]) to analyse pharmaceutical company reports, drawing attention to how companies are reporting and aiming to reduce emissions. We have added further information about this approach to the Methods section. The focus here is on providing an overview of what pharmaceutical companies report they are doing, thereby beginning to address a major gap in the current literature on an important component of the health system supply chain. There are good reasons to believe that annual reports are a reliable source of information on company plans and progress, given that they are investor materials, are heavily regulated and audited, with those misrepresenting information to the stock market faced with penalties. However, we note the limitations of such an approach and have made this clearer in the paper.

We agree that there is significant work still to do to examine current practices in the pharmaceutical industry (i.e., not just what they report, but what they do). This is outside of the scope of this research paper. We thank the reviewer for the suggestion of conducting interviews with employees of pharmaceutical companies, which one author intends to pursue as part of her doctoral studies. We have added further recommendations for future research as per your comments.

The case: the introduction contains interesting references suggesting that the pharmaceutical sector is an exemplar case to study, but adding a section about that topic in the methodology section could add more evidences justifying why this sector is adequate to that research.

Thank you for your comment. We chose the pharmaceutical sector to study because it is very topical to understand what climate change targets and GHG emissions they are reporting and how they plan on reducing them in light of health systems globally increasingly making commitments to net zero along both their operations and supply chain (including the pharmaceutical industry)[4]. We also note the gap in literature on this topic and the importance of understanding the nuances present in reporting and reducing emissions in the pharmaceutical industry that are difficult to generalise from literature on other industries. We have added justification for choosing this sector in the Introduction section.

Scope for the research/ unit of analysis: The perimeter of the research is very broad, as it includes “scope 1, scope 2 and scope 3” emissions. By involving scope 3, research should incorporate also the entire supply chain, at least the upstream supply chain and collect data from the suppliers themselves.

Our starting point was understanding GHG emissions of health systems and how they will meet their climate change targets (e.g., NHSs commitment to net zero along their supply chain by 2045). We set out to explore an element of this by focusing on a substantial contributor of health systems scope 3 emissions – the pharmaceutical supply chain. The global carbon disclosure system is dependent on organizations taking responsibility for the emissions of their suppliers; each organization needs to calculate and report the emissions attributable to them. This necessarily requires some assumptions but creates a workable system. The alternative of contacting the suppliers and then their suppliers would be challenging – for example, approaching an airline to ask what proportion of the flights made by an accountancy firm were attributable to a specific pharmaceutical company. Ultimately, collecting data directly from pharmaceutical suppliers is beyond the scope of this research and given the challenges of accessing such (often proprietary) data, is not feasible.

Data analysis. Several sentences are not clear, or mention concepts which are not exposed in the paper. For instance: “Reported scope 1, 2, and 3 GHG emission values for the companies were documented and converted to the same units (metric tonnes carbon dioxide equivalents - MtCO2e) to ensure conformity.” Which conformity is this about? What kind of data are reported? Where? How?

Thank you for picking up on this. We initially had a table that presented the absolute values of reported GHG emissions for scope 1, 2, and 3 converted to the same units. However, we decided that it would be easier for readers to visualize the change in emission values from the various companies’ baseline years of reporting as a bar graph (Figure 2) rather than being presented with a table of numbers. We moved the table to supplementary material and so the sentence you picked up on refers to data present in Supplementary Table S4. To make this clear, we have re-worded the sentence so that readers do not expect this data within the actual paper, and referred to Supplementary Table S4 in the Results section.

Results are questionable, for the reasons mentioned above (nature of the data collected).

Research based only on secondary data does not allow one to conclude the vast majority have active commitments to reducing their GHG emissions. This conclusion is only based on advertised position of each of these firms.

The focus of our research was to describe what targets, emissions, and reduction strategies companies are reporting in their company reports. Our results, therefore, address our research questions and outline what pharmaceutical companies report with regard to the targets they have publicized, the emissions they report, and the steps they report that they are taking to reduce emissions. Given the self-reported nature of the GHG emission disclosure system (as outlined above) we stand by our methodology of gaining this information via content analysis of public documents. We have clarified in the text that targets and progress are what companies are publishing and/or reporting. Nevertheless, as explained above, we believe that published targets included in annual reports can indeed be considered public commitments. While we recognise that there is further work to be done here, and fully acknowledge that what companies report is not necessarily the same as what they are doing to progress this agenda, we are clear about what this research set out to do and present the results accordingly. We have not, therefore, made changes to the results. However, we have (a) changed the title to reflect that this research is about ‘reported’ targets, emissions and strategies, not necessarily actions; (b) extended the discussion to make clearer that this is about reported actions, and may not therefore reflect subsequent action; (c) cited relevant literature relating to greenwashing as a possible explanation for positive reporting, (d) made it clear that there are limitations to analysis of secondary data in giving access solely to what companies report, (e) added the terms ‘reported’, ‘publicized’, or ‘published’ throughout the paper when referring to data sources, and (f) added to the section outlining a future research agenda in this field. 

Same for the following suggestion: “Third-party assurance is also important to give credibility to company’s reporting data, and it was encouraging to note that all companies reported that their data had undergone auditing”.

We agree. We have changed this to simply “…all companies reported that their data had undergone auditing”, avoiding any claims about the accuracy of the audits as that is outside the scope of this research.

One of the interesting finding in this research is the following: “We found that the total reported emissions intensity of the 20 pharmaceutical companies in this study was 151MT CO2e/$M of revenue.” But it is like comparing apples and carrots! You cannot sum up all these numbers to get only one single result. What are the calculation algorithms behind each number collected from each company? Example: in the GHG protocol, spend data are mentioned in monetary or quantity-based values, and are combined with globally established databases to calculate the CO2 emissions for the scope 3. The input of spend is therefore necessary to bridge with CO2e/$M, but we haven’t seen these figures in the paper. Nowadays, the emissions data usually come from a combination of licensed and open data sources including EcoInvent and Exiobase. Ecoinvent data is used for quantity-based values and Exiobase for monetary values. Have you collected the Exiobase numbers, to be able to make your conclusions? Also, CO2 analytics should take inflation and currency rates into account to match with the databases to increase accuracy. This has not been done here.

Thank you for your comment. We noted that two other published papers (Ray et.al., and Belkhir et.al.) performed similar calculations of CO2e/$M without doing the above. We made such calculation on the assumption that companies all reported using the GHG Protocol to determine their GHG emissions (and therefore hopefully used similar methods). However, we note your concerns with this and have opted to remove this result as it will not be possible to obtain proprietary information from pharmaceutical companies on their aggregated data and supplier spending to perform our own calculations.

Last, in addition to secondary data, CO2 analytics should provide supplier-specific information directly from suppliers. As the unit of analysis is unclear (supposedly pharma companies or the entire sector, or only one firm), this missing information reduces the power of the conclusions.

As stated above, gathering data on suppliers CO2 emissions is outside the scope of this research. We are already collating information from key suppliers of health systems.

Last, the secondary data provided for all CO2 emissions (Exiobase, etc ) are also based on assumptions and averages for a specific purchase category and supplier location. You would perfectly understand that data provided by annual reports are not valid, because all the firms considered could change their supply base from one day to another, making the calculation obsolete. You should take this into account and therefore pay closer attention to the accuracy of the underlying calculations to limit further bias. Purchasing categories and emission categories are linked together, so there is a chance the best match is not achieved! Another limitation is the challenge of receiving perfect unit of measurement conversions, in which case the database Exiobase is prioritised. Finally, the accuracy of the calculations may differ depending on the underlying spend data, which is company specific.

We refer the reviewer to the comments made above. We agree that reported data obtained is subject to limitations. We make it clearer in the paper that the data reported by companies is the best available data that we currently have, also that our focus is not an in-depth analysis of how companies calculated their emissions, but rather to describe what emissions they are reporting. Additionally, we have added a paragraph to the limitations section where we note the intrinsic limitations of measuring and reporting emissions.

Reviewer 3 Report

as attached

Author Response

REVIEWER 3

The authors claim that this study is the first one to assess the climate change commitments and emission reduction strategies of leading pharmaceutical companies. In fact, this study is not only the first one to assess the emission reduction strategies for climate change commitments but also the first study employs the content analysis. The authors can add this contribution and distinct to at the end of the introduction instead of in the section of Discussion.

Thank you for your positive comment and support of our overarching approach. In light of feedback from other reviewers, we have elected to extend our Methods section and provide further detail about our approach and use of content analysis there.  

The authors suggest that it would be beneficial if stakeholders, including academics, policy makers, and industry, could collaborate to develop a standard definition for carbon neutral and net zero and what each commitment involves. There is clear definition both for both terms. Since the authors have such suggestion it means the authors are not clear about the meaning of both terms. The definition of carbon neutral is a mean of achieving net zero. It can be offset by tree growing or using emission finance to purchase emission certificate or invest project on renewable energy or carbon removal. Thus, it is not necessary with actual carbon emissions. Carbon neutral includes only CO2. Moreover, carbon neutral includes only scope 1 and scope 2 emission only. Net zero then includes scope 1, scope 2, and scope 3. Carbon neutral is a short-term objective and net zero is a long-term objective. If the authors have such suggestion it means the authors complete the analysis for this study under the unclear definitions of these terms. I will suggest the authors to clear up the full manuscript under the definitions of these two terms provided above.

Thank you for this comment. We are aware that definitions of ‘carbon neutral’ and ‘net zero’ exist, and conducted the study in line with those. We now make this clearer in the Introduction, which now includes a section on terminology. We have clarified that what is needed is not standard definitions (which already exist) but standard operationalizations of those definitions. To make this clearer, we have rephrased this sentence as follows: “It would be beneficial if stakeholders, including academics, policymakers, and industry, could collaborate to agree on how to operationalize climate change targets (and the terms used to refer to these targets) and what each involves.” Line: 606-609.

References:

  1. Downe‐Wamboldt B. Content analysis: Method, applications, and issues. Health Care for Women International. 1992;13(3):313-21.
  2. Bengtsson M. How to plan and perform a qualitative study using content analysis. NursingPlus Open. 2016;2:8-14.
  3. Milne MJ, Adler RW. Exploring the reliability of social and environmental disclosures content analysis. Accounting, Auditing & Accountability Journal. 1999;12(2):237-56.
  4. Wise J. COP26: Fifty countries commit to climate resilient and low carbon health systems. BMJ. 2021;375:n2734.